# Characterization of Cationic Modified Short Linear Glucan and Fabrication of Complex Nanoparticles with Low and High Methoxy Pectin

**DOI:** 10.3390/foods10102509

**Published:** 2021-10-19

**Authors:** Wenhui Li, Ying Yu, Jielong Peng, Ziyang Dai, Jinhong Wu, Zhengwu Wang, Huiyun Chen

**Affiliations:** 1Department of Food Science and Engineering, School of Agriculture and Biology, Shanghai Jiao Tong University, Shanghai 200240, China; liwenhui3781692@sjtu.edu.cn (W.L.); diamond-yy@sjtu.edu.cn (Y.Y.); pjl0402@sjtu.edu.cn (J.P.); dzy1516289438@sjtu.edu.cn (Z.D.); zhengwuwang@sjtu.edu.cn (Z.W.); 2Institute of Agricultural Product Processing Research, Ningbo Academy of Agricultural Science, NO. 19 Dehou Street, Yinzhou District, Ningbo 315040, China; chhyun@163.com

**Keywords:** cationization short linear glucan, pectin, encapsulation efficiency, cumulative release

## Abstract

In this study, we chemically modified the short linear glucan (SLG) using the 3-chloro-2-hydroxypropyl trimethylammonium chloride to introduce a positive surface charge via cationization (CSLG). We then prepared CSLG-based binary nanocomplex particles through electrostatic interactions with low and high methoxyl pectin. The two new types of binary nanocomplex were comprehensively characterized. It was found that the nanocomplex particles showed a spherical shape with the particle size of <700 nm, smooth surface, homogeneous distribution, and negative surface charge. Fourier transform infrared spectroscopy (FTIR) revealed that the driving forces to form nanocomplex were primarily electrostatic interactions and hydrogen bonding. In addition, increasing the CSLG concentration in the nanocomplex significantly enhanced both thermal stability and digestive stability. By comparing the two complex nanoparticles, the HMP-CSLG has a larger particle size and better stability under the GI condition due to the high content of the methoxy group. Additionally, the HMP-CSLG nanoparticle has a higher encapsulation efficiency and slower release rate under simulated gastrointestinal fluid for tangeretin compared with the LMP-CSLG. These results provide new insights into designing the CSLG-based nanocomplex as a potential oral delivery system for nutraceuticals or active ingredients.

## 1. Introduction

Nanotechnology has emerged as a promising way to revolutionize food and nutrition by introducing a wide range of nanoscale systems, including but not limited to nutrient delivery, nanosensors, and nanoscale antimicrobials [1,2,3]. Among these systems, the nanoparticle has been widely used in many fields because it can be fabricated by natural biomaterials with low-cost techniques under facile conditions. The biomaterials-based nanoscale systems have recently attracted increasing attention in delivering bioactive compounds due to their biodegradability, biocompatibility, non-antigenicity, renewability, and extraordinary binding capacity of bioactive ingredients [3,4]. Notably, polyelectrolyte nanocapsules, a typical nanoscale delivery system, which is prepared layer by layer electrostatically through alternate interactions of positively and oppositely charged biomacromolecules, are promising candidates for bioactive molecule-controlled release systems [5]. The polyelectrolyte complexes could protect the bioactivities of the ingredients by maintaining their stable state under a harsh gastrointestinal tract environment [6].

Starch is an abundant natural resource, which has been applied in various fields due to its desirable functional attributes, such as non-toxicity, inexpensiveness, biodegradability, biocompatibility, and ease of chemical modification by chemical or enzymatic processes [7]. However, native starch has some limitations, such as high viscosity, limited solubility, and dispersibility [8]. Several methods have been applied to modify the native starch (including the physical, chemical, enzymatical, and genetical methods) to investigate their potential use [9]. Cation modification is an important modification, which can not only be applied to evaluate the functionality of polysaccharides but to introduce a positive charge to the surface of the polysaccharides for the formation of the polyelectrolyte nanocapsule [10]. Luan et al. reported that a quaternization reaction could significantly improve the antioxidant activity of chitosan [11]. Cationized starch is an important derivative of starch, widely employed as a wet-end chemical in the paper industry and adsorbent in raw or wastewater purification [12,13]. However, there are few studies using the cationized starch to fabricate the nanocomplex with another electrolyte polysaccharide.

The short linear glucan (SLG) is an enzymatically (pullulanase or isoamylase) modified starch, which has been widely employed in the food and pharmaceutical industries due to its biodegradability, biocompatibility, and non-toxicity [14]. For instance, it is a good tablet matrix to extend the duration of drug release [15]. Ji et al. used the SLG to synthesize nanocomposite with proanthocyanidins for the oral delivery of insulin [16]. Li et al. used the SLG and protein to prepare the SLG/protein hybrid nanoparticles, which could significantly improve the rheological properties of starch gel [17]. Moreover, SLG has been modified by acetylation, and the swelling power and freeze-thaw stability of the SLG has been found to be significantly improved [18]. Chang et al. used OSA-modified SLG to prepare the starch vesicles for loading hydrophilic functional ingredients [19]. Therefore, we assume that the SLG could be cationized, and the cationized SLG can be applied to develop a novel delivery system for the bioactive compound.

Pectin is a typical polysaccharide, and it has been widely applied in the food industry as a gelling or stabilizing agent [20]. According to the degree of methylation (DM), pectin can be classified into high methoxyl pectin and low methoxyl pectin. In addition, pectin is an anionic polysaccharide and can be used as polymeric multilayer coatings to fabricate homogeneous and re-dispersible solid nanoparticles, which can maintain stability in gastrointestinal fluid [21]. Despite the extensive use of pectin in the biomedical field, there are few reports concerning pectin’s use as a polyelectrolyte layer coating on the surface of the starch derivatives to develop a delivery system for the bioactive compounds.

In this work, we modified the short linear glucan by reaction with CHPTAC to obtain the cationic short linear glucan (CSLG) and prepared the polyelectrolyte nanoparticles. The CSLG was employed for constructing layers with positively charged groups, while high methoxyl pectin and low methoxyl pectin were used as oppositely charged polymer species. We comprehensively characterized the obtained complex nanoparticles with different CSLG concentrations and used them as carriers to encapsulate tangeretin. The encapsulation and release characteristics of tangeretin in nanocomplex under simulated gastrointestinal fluids conditions were investigated.

## 2. Materials and Methods

### 2.1. Material

Waxy maize starch was obtained from Gaofeng Starch Technologies Co., Ltd. (Suzhou, China). Low and high methoxyl pectin and (3-Chloro-2-hydroxypropyl) trimethylammonium chloride (CHPTAC) were purchased from Shanghai Yuanye Biotechnology Co., Ltd. (Shanghai, China). Pullulanase (4461.6 NPUN/g) was purchased from Novozymes Investment Co., Ltd. (Beijing, China). Pepsin and pancreatin were purchased from Sigma–Aldrich Chemical Company (Shanghai, China). All chemicals were of chromatographic or analytical quality and were purchased from Sinopharm Chemical Reagent (Shanghai, China).

### 2.2. Preparation of the CSLG

The CSLG was prepared according to the method described by Qin et al. [22]. Briefly, waxy corn starch was fully gelatinized (100 °C for 30 min) and debranched using pullulanase at 58 °C for 24 h. Afterward, the mixture solution was centrifuged, and the supernatant was precipitated and washed to neutral. In the end, the sediments were lyophilized to obtain dried SLG. The CSLG was prepared according to the method described by Liu et al. [10], with a slight modification. The SLG (2 g) was dispersed in 100 mL of deionized water and sonicated for 10 min, and the pH was adjusted to 12 with 3 M of sodium hydroxide solution at 40 °C. After that, the CHPTAC (1 mL) was added to react by keeping the pH value at 12. After a 4-h etherification reaction, the pH of the dispersion liquid was adjusted to 7 with 1 M of hydrochloric acid, followed by the centrifugation of the suspension, and the precipitate was washed with 95% ethanol and lyophilized to obtain the CSLG. The same procedure was used to prepare other doses of CHPTAC (2 and 3 mL). The SLG modified by CHPTAC with different contents (1 mL, 2 mL and 3 mL) was named CSLG-1, CSLG-2, and CSLG-3, respectively. The chemical structure of the CSLG and the chemical reaction path of cationic modification is shown in Figure 1.

### 2.3. Characterization of CSLG

The degree of substitution (DS) of the CSLG was measured by determining the nitrogen content of the CSLG with a Kjeldahl method [23]. Briefly, CSLG (1 g), CuSO_4_·5H_2_O (0.7179 g), and K_2_SO_4_ (6.2821 g) were mixed and hydrolyzed in the concentrated sulfuric acid (12 mL) at 420 °C for 1 h. When the digestive liquid turned clear, it was taken out and cooled to room temperature before measurement on a Kjeltec-2300 autoanalyzer. The DS was calculated using the following equation:DS = (162 × N)/(1400 − 151.5 × N)(1)
where N is the amount of nitrogen measured by using the Kjeldahl method, 162 is the average molecular weight of the anhydroglucose unit, 1400 is 10 times the value of the atomic weight of nitrogen, and 151.5 is the molecular weight of CHPTAC without chloride group.

The structure analysis of the SLG and CSLG were performed using a proton nuclear magnetic resonance (^1^H NMR) spectrometer (AVANCE NEO 600 MHz, Bruker, Switzerland), and the chemical shifts were expressed in ppm. The chemical structures of SLG and CSLG were measured by a Fourier transform infrared (Thermo Fisher Co., Waltham, Massachusetts, USA) spectrum, and the FTIR spectra were collected in the range of 500–4000 cm^−1^. The XRD spectra of SLG and CSLG were measured by a Bruker-AXS D8 ADVANCE powder diffractometer. In addition, a Zetasizer Nano Series measured the zeta potential of SLG and CSLG at 25 °C.

### 2.4. Preparation of the CSLG-Based Nanocomplex

The nanocomplex was prepared according to the method of Liu et al. [10]. Firstly, the low methoxyl pectin (LMP) and high methoxyl pectin (HMP) solutions (0.5 mg/mL) solution were stirred overnight to induce the complete hydration and the CSLG solution (0.5, 1.0, 2.0, and 5.0 mg/mL) were freshly prepared. For the formation of nanocomplex particles, the CSLG was dispersed into the boiling water for 30 min, and then, it was added dropwise to an equal volume of LMP solution at 50 °C with constant stirring at 500 rpm. The same was done with an equal volume of HMP solution. The mixture was incubated for 2 h and then centrifuged at 10,000× *g* for 30 min. Finally, the precipitate was washed and lyophilized to obtain the dried nanoparticles. The nanocomplex particles formed by complexing the LMP with different mass ratios (0.5, 1.0, 2.0 and 5.0 mg/mL) CMSLG were name as LMP-CMSLG 0.5, LMP-CMSLG 1.0, LMP-CMSLG 2.0 and LMP-CMSLG 5.0. Similarly, The nanocomplex particles formed by complexing the HMP with different mass ratios (0.5, 1.0, 2.0 and 5.0 mg/mL) CMSLG were name as HMP-CMSLG 0.5, HMP-CMSLG 1.0, HMP-CMSLG 2.0 and HMP-CMSLG 5.0.

### 2.5. Particle Size, Polymer Sispersity Index (PDI), and Zeta Potential

Both average droplet size and zeta potential of nanocomplex particles were determined by a Particle Size and Zeta Potential Analyzer (NanoBrook Omni, Brookhaven Instruments, MS, USA).

### 2.6. Transmission Electron Microscopy (TEM) Imaging

The morphology of the nanocomplex particles was observed by a TEM (Talos L120C G2, Thermo Scientific, MA, USA) at an accelerating voltage of 120 kV. A droplet sample dispersion (0.1% *w/v*) was dropped in a carbon-coated copper grid (300 meshes) and then lyophilized for 10 h to obtain the dry samples for the TEM observation.

### 2.7. Fourier Transform Infrared Spectroscopy (FTIR)

The FTIR spectra of the nanocomplex particles were measured by a Fourier transform infrared spectroscopy (Thermo Fisher Co., Waltham, Massachusetts, USA) in the range of 500–4000 cm^−1^.

### 2.8. Differential Scanning Calorimetry (DSC)

The thermal properties of the nanocomplex particles were determined by a DSC (TA 2910, TA Instruments, Wilmington, DE, USA). Samples (5.0–10.0 mg) were placed inside aluminum pans and sealed, followed by heating from 25 to 250 °C at a heating rate of 10 °C/min.

### 2.9. X-ray Diffractogram (XRD

The crystalline structure of all nanocomplex particles was determined by a Bruker-AXS D8 ADVANCE powder diffractometer using Cu-Kα radiation (wavelength *λ* = 1.54 nm) at 40 kV and 30 mA over the angular range 2*θ* = 5–40° and a step time of 2.0 s. The relative crystallinity was calculated using the instrument software (Jade 6.5, Materials Data, Inc., Livermore, CA, USA).

### 2.10. Nanoparticle Stability in Simulated Gastrointestinal (GI) Fluids

The simulated GI fluids, including the Simulated gastric fluid (SGF, 0.32% pepsin, pH 2) and simulated intestinal fluid (SIF, 1% pancreatin, pH 7), were prepared according to the method described by Berardi et al. [24]. Nanocomplex particles dispersion (5 mg mL^−1^) solution was diluted (1:9) with either SGF or SIF and incubated for 2 h (in SGF) or 4 h (in SIF). The incubation temperature was set as 37 °C. The particle size and zeta potential of samples were measured by Particle Size and Zeta Potential Analyzers (NanoBrook Omni, Brookhaven Instruments, MS, USA).

### 2.11. Encapsulation of Tangeretin (TAN)

The TAN solution was freshly prepared by dissolving 100 mg of TAN into 20 mL of ethanol. Then, the TAN solution was pipetted into the CSLG solution at 50 °C under constant stirring for 120 min, followed by the same procedure to prepare complex nanoparticles as described in Section 2.4. Then, the TAN-loaded nanocomplex particles suspension was centrifuged at 10,000× *g* for 30 min, and the supernatant was used to calculate the encapsulation efficiency (EE) of TAN. A UV–vis spectrophotometer (Thermo Scientific, Madison, WI, USA) was used to measure the absorbance of TAN in the supernatant at the wavelength of 328 nm. The EE of the TAN was calculated according to the following equations:(2)$$\mathrm{EE}=\frac{\mathrm{Initial}\;\mathrm{amount}\;\mathrm{of}\;\mathrm{TAN}-\;\mathrm{amount}\;\mathrm{of}\;\mathrm{free}\;\mathrm{TAN}\;\mathrm{in}\;\mathrm{supernatant}}{\mathrm{Initial}\;\mathrm{amount}\;\mathrm{of}\;\mathrm{TAN}}\;\times \;100$$

### 2.12. The Release Properties of the TAN in the Nanocomplex Particle

The release properties of the TAN from the complex nanoparticles were carried out with an equilibrium dialysis method. Firstly, 5 mL of TAN-loaded nanocomplex particles were placed into a dialysis bag (with 10 kDa molecular cutoff) and dialyzed against 30 mL of SGF (pH 2) release medium and incubated at 37 °C for 2 h. Afterward, the dialysis bag was transferred to another flask containing 30 mL of SIF (pH 7) release medium and incubated for 4 h. All the incubation temperatures were 37 °C. The release medium (1 mL) was collected and replaced with a fresh medium at various times. The concentration of the TAN was measured by a UV−vis spectroscopy as described in Section 2.10. Free TAN was used as a control.

### 2.13. Statistics

Each test was carried out in triplicate, and all experimental results are shown as means ± standard deviations. Data were analyzed by one-way analysis of variance (ANOVA), followed by Tukey’s test using SPSS 17.0 Statistical Software Program (SPSS, Chicago, IL, USA). *p* < 0.05 was considered statistically significant.

## 3. Result and Discussion

### 3.1. Characteration of the CSLG

The DS represented the average number of hydroxyl groups substituted in one anhydrous glucose unit of SLG [23]. Table 1 showed the DS and zeta potential of SLG and CSLG. Compared with SLG, the DS of the CSLG significantly increased. Moreover, as the amount of the etherifying agent increased from 1 mL to 3 mL, the DS increased significantly from 0.36 to 0.94, which indicated the etherification reaction happened successfully. Xu et al. [25]. found that the zeta potentials of quaternized β-chitin derivatives increased from +25.4 to +44.1 mV with the degree of quaternization increasing from 0.18 to 0.43.

The introduction of the cationic group of the CSLG was further confirmed by ^1^H NMR, and the spectrum is shown in Figure 2A. For all the samples, the peak near 3.6 ppm was attributed to the hydroxyl group [26]. Compared to the unmodified SLG, the CSLG present a new peak at near 3.14 ppm, which was caused by the H of the—N^+^(CH_3_)^3^ introduced into the backbone of the SLG [25]. Moreover, the peak intensity increased gradually as the DS of CSLG increased. The results suggest that the—N^+^(CH_3_)^3^ groups were successfully grafted onto the SLG.

The FTIR spectra of the SLG and CSLG was showed in Figure 2B. The SLG and CSLG showed several similar characteristic peaks, including a strong vibration at 3343 cm^−1^ for O-H asymmetric stretching and 2930 cm^−1^ for C-H bonding. Meanwhile, there are three characteristic peaks in the range of 1000–1200 cm^−1^ for C−O stretching vibrations. Compared with the SLG, the CSLG showed a new peak at 1482 cm^−1^, which ascribed to the C−N stretching vibration [27], and the intensity of the peak increased as the etherifying agent increased from 1 mL to 3 mL. The result also implied that the cationic group was successfully introduced to the SLG by the etherification reaction following the results with ^1^H NMR.

The XRD spectra of the SLG and CSLG were presented in Figure 2C. The XRD spectra of SLG and CSLG all showed an “A” type crystallinity and had four main diffraction peaks at 15°, 17°, 18° and 23.5°. The CSLG showed a similar crystal type with SLG but a lower peak intensity, and the relative crystallinity was decreased from 38.6% to 17.3% as the etherifying agent increased from 1 mL to 3 mL, which suggested that the cationization could damage the crystalline structures of the SLG. A similar result has been found by other researchers [10,28].

The zeta potential of the SLG and CSLG is also present in Table 1. The zeta potential of SLG is −12.13 mV, consistent with the previous studies [10]. As expected, the zeta potential of the CSLG increased significantly and became positive, reaching +34.52 mV when the DS was 0.94. The phenomenon suggested that the cationic group has been grafted onto the SLG.

### 3.2. Characterization of Nanocomplex

Based on the above results, the CSLG of DS 0.94 was selected for preparing the nanoparticles with LMP and HMP. The particle size, PDI, and zeta potential of the various complex nanoparticles are presented in Table 2. When the CSLG concentration was 0.5 mg/mL, the particle size of the LMP-CSLG nanocomplex is 488.85 nm with a PDI of 0.203, and it was significantly increased to 629.23 nm as the CSLG concentration increased to 5.0 mg/mL (Table 2) with an increased PDI of 0.300. Similar trends were found for the HMP-CSLG. However, the HMP-CSLG has a higher particle size and PDI compared with the LMP-CSLG at the same concentration of CSLG (Table 2). During the formation of the complex nanoparticles, the cationic groups of CSLG interacted with the negatively charged carboxyl groups of pectin, leading to the pectin deposit onto CSLG through electrostatic attraction. Thus, electrostatic attractions played an important role in the formation of binary nanocomplex particles. The LMP-CSLG has a lower zeta potential (Table 2) due to the lower methoxyl group compared with HMP [29], and it causes a higher electrostatic attraction when interacting with the CSLG. The higher electrostatic attraction prompt the formation of a compact structure of the nanoparticle with a smaller particle size and a lower PDI value (Table 2). From another point, the high methoxyl group of the HMP also leads to higher viscosity of the complex nanoparticles, which could cause the aggregate of the particles, leading to a higher particle size and PDI [7].

### 3.3. TEM

The TEM images of the LMP-CSLG and HMP-CSLG nanocomplex particles are presented in Figure 3. The LMP-CSLG (Figure 3a–d) exhibited near-spherical shapes in general with a size around 300–500 nm, which was slightly smaller than the data from particle size analyzer measurement (Table 1). This is because the particle size from TEM is obtained from dehydrated particles that have shrinkage in the dimension at their dry status under TEM observation. Moreover, as the CSLG concentration increased, the particle size of the two nanocomplex particles increased gradually and the number of the small particle decreased, which was consistent with the result obtained from Section 3.2. The strong electrostatic attraction caused a compact and uniform structure of the LMP-CSLG complex nanoparticle [30]. However, as the CSLG increased in the system, a large number of CSLG adsorbed on the surface of the nanoparticle, leading to the increase in the particle size. For the HMP-CSLG complex nanoparticle (Figure 3e–h), it showed an irregular shape with a larger particle size compared with the LMP-CSLG nanoparticles. This phenomenon also can be explained by the higher viscosity of the HMP, leading to the particle aggregate in the system.

### 3.4. FT-IR

The FT-IR spectra of individual LMP and HMP and the nanocomplex of LMP-CSLG and HMP-CSLG are presented in Figure 4A,B. The LMP and HMP showed similar spectra; it has two peaks at 3384 cm^−1^, and 2930 cm^−1^, originating from O-H and C–H stretching vibration, respectively [31]. The peaks at around 1740 and 1630 cm^−1^ were ascribed to the esterified carbonyl (C=O) and carboxylic acid ion (COO-) stretching vibrations. Notably, after the formation of the complex nanoparticles, the characteristic peaks exhibited several major changes, indicating the strong interactions between the cationic groups from CSLG and an anionic group from pectin and forming the electrostatic complex. First, the characteristic peak of O-H stretching vibrations (3384 cm^−1^) became sharper and broader, indicating that hydrogen bonding was one of the binding forces in forming CSLG/pectin complexes [32]. Additionally, the redshift of the vibrational peaks for C=O bonds (1740 cm^−1^) was phenomenal and accompanied by a reduction in their intensities as the CSLG concentration increased gradually (Figure 4A,B). These results evidenced the existence of strong electrostatic interactions among the CSLG and the carboxylate groups of pectin in the CSLG-pectin nanocomplex particle. There are no significant differences in the FTIR spectra between the LMP-CSLG and HMP-CSLG nanoparticles.

### 3.5. DSC

The thermal property of the individual biopolymer and the nanocomplex particles was carried out by DSC (Figure 4C,D). The smooth thermograms indicate the inherent amorphous structures of native biopolymer and the binary complexes [33]. As shown in Figure 4C, the DSC profile of LMP displayed one endothermic transition at about 145 °C with an enthalpy of 138.56 J/g (Appendix A), consistent with the previous report [34]. However, after the formation of the binary nanoparticles, the endothermic transition of LMP-CSLG nanoparticle sharply decrease to 90 °C, significantly lower than the endothermic transition of native LMP. The decrease might be attributed to the lower water-holding capacity and increased hydrogen bonding of nanocomplex [35], indicating that driving forces to form nanocomplex were primarily electrostatic interactions and hydrogen bonding. On the other hand, with the CSLG concentration increased from 0.5 mg/mL to 5.0 mg/mL, the endothermic peaks widened, and the enthalpies increased from 102.54 to 129.63 J/g. This result indicated that the addition of the CSLG enhanced the thermal stability of the LMP-CSLG nanocomplex by increasing the hydrogen bonding between CSLG and LMP [36]. Similarly, HMP presented an endothermic peak at about 150 °C, which was decreased to 92 °C after the formation of the nanocomplex particles (Figure 4B). As shown in Figure 4C, as the CSLG concentration increased gradually, the enthalpies of the LMP-CSLG increased gradually; however, there is no regularity of the enthalpy changes of the HMP-CSLG. This is because the aggregate phenomenon of the HMP-CSLG is heavy due to the high viscosity of HMP.

### 3.6. XRD

The XRD patterns and the relative crystallinities of the LMP, HMP, and their nanocomplex particles are shown in Figure 4E,F. The LMP showed two broad-type diffraction peaks around 12° and 22°, respectively, indicating that the samples had the widely accepted crystalline structure of pectin [37]. After forming the LMP-CSLG nanocomplex particle, the intensity of the characteristic peaks decreased significantly, and only a flat peak can be observed at 2*θ* equal to 13°. However, with the increase in the CSLG concentration, two peaks at 2*θ* equal to 17° and 23° appeared, indicating a typical B-type crystalline, and the peak intensity increased gradually. These results indicated that the formation of the complex nanoparticles successfully occurred, and the hydrogen bonding was the main driving force. As the CSLG increased, the interaction between CSLG and pectin enhanced, leading to more hydrogen bonding [38]. As the crystalline structure of LMP-CSLG was dominantly realized by forming intermolecular hydrogen bonds, increasing the content of hydrogen bonds could result in more ordered and exquisite crystalline structures in LMP-CSLG complex nanoparticles [39]. As Figure 4F showed, the HMP-CSLG has a similar XRD pattern but a higher relative crystallinity compared with LMP-CSLG nanoparticles.

### 3.7. GI Stability of Nanocomplex Particle

The colloidal stability of the different nanocomplex particles under simulated GI conditions was evaluated, and the changes of particle size and zeta potential are presented in Figure 5. In the SGF, the two types of nanocomplex showed significant instability with increasing particle size (Figure 5A,B). For example, the particle size of the LMP-CSLG nanocomplex particle at a CSLG concentration of 0.5 mg/mL is 488.8 nm, and it increased to 585.2 nm after being digested in the SGF. This is because the extremely low pH (pH 2) causes the biopolymers to become highly protonated, weakening the interaction between CSLG and pectin, leading to the pectin accumulate on the surface of the nanocomplex particle [30]. Chang et al. [40] has demonstrated that pectin coating provided excellent stability to NaCas/zein nanoparticles under simulated gastrointestinal conditions.

In contrast, after incubation in SIF, the particle size of all nanocomplex particles decreased significantly (Figure 5A,B). This is because the complex nanoparticles were partially hydrolyzed by pancreatin [10]. It must be noted that the increased CSLG concentration of the nanocomplex mitigated the above change. The abundant CSLG free molecules could adsorb some digestive enzymes, reducing the effects of digestive enzymes on complex nanoparticles. In addition, comparing the LMP-CSLG complex particles, the particle size of HMP-CSLG complex particles showed small changes at low CSLG concentration due to the high viscosity of the HMP. On the other hand, it has been reported that the methoxy group can interact with the digestive enzyme via hydrogen bonding, which can decrease the digested rate of the HMP-CSLG complex nanoparticle [41].

As shown in Figure 5C,D, both the SGF and SIF condition could increase the zeta potential of the nanocomplex particles. Especially under the condition of SIF, the nanocomplex have the highest zeta potential. This phenomenon is due to the protonation of the cationic group in the nanocomplex at a very low pH, which leads to the increased magnitude of the positive zeta potential of the nanocomplex [42].

### 3.8. Encapsulation and Controlled Release of TAN

To study the encapsulation and delivery capability, TAN was used as a model bioactive compound and loaded into two different types of nanocomplex with four different CSLG concentrations. The EE of the TAN in the nanocomplex is shown in Figure 6A. Interestingly, the HMP-CSLG nanocomplex had a higher EE compared with the LMP-CSLG complex particles. For example, when the CSLG concentration was 0.5 mg/mL, the HMP-CSLG have an EE of 72.18%, which was significantly higher than the LMP-CSLG (61.41%). As the CSLG concentration increased, both types of nanoparticles exhibited an increase in the EE for the TAN, and the difference in EE between the two nanoparticles gradually narrowed (Figure 6). The higher EE of the HMP-CSLG is attributed to the high content of the methoxy group in HMP, which helps to adsorb more TAN in the nanoparticles [40]. The increased EE of the nanoparticles by increasing the CSLG concentration could be explained by the increased available space within the nanoparticles for TAN to be entrapped due to because CSLG is a superior host for hydrophilic molecules [43]. As the CSLG increased in the nanoparticles, the effects of the content of the methoxy group reduced. In addition, as shown in Figure 6B, both types of nanocomplex particle exhibited sustained release of TAN under SGF and SIF, with about 70–80% cumulative release at the end of the incubation period, which was significantly lower than the free TAN in the simulated GI (Figure 6B). The free TAN showed a burst and zero-order release pattern, with a fast diffusion rate in SGF. Within the first 2 h, more than 90% of TAN diffused into the release medium from the inside dialysis membrane due to the sufficient osmotic pressure, and the rest of the TAN diffused into the SIF in the following incubation. In contrast, the two types of nanocomplexes exhibited a much slower and sustained release of TAN throughout the incubation period. Moreover, the release rate of TAN in SIF was slightly higher than that in SGF. The presence of pancreatin might decrease the protection capacity of the pectin as a result of the digestion of the nanocomplex. Liu et al. [10] has reported that for both cationic starch/κ-carrageenan and cationic starch/carboxymethyl chitosan nanoparticles, the release rate of EGCG in SIF was slightly higher than that in SGF. It is worth noting that the HMP-CSLG nanocomplex showed a slower cumulative release during the whole incubation compared with the LMP-CSLG nanocomplex. This can be explained from two aspects: (1) the HMP having a high viscosity as the outmost coating layer provided excellent protection of CSLG from digestive degradation and thus greatly reduced the release rate of curcumin [40], (2) or the high content of the methoxy group could enhance the interaction between the CSLG and pectin via the hydrogen bonding, leading to the TAN being more difficult to release from the complex nanoparticles [44].

## 4. Conclusions

In this work, the SLG was modified by the CHPTAC and introduced the quaternary ammonium groups onto the SLG through etherification reaction to obtain the CSLG. The results of DS and FT-IR of the CSLG showed that the cationic group was successfully grafted onto the backbone of SLG, and the XRD result revealed that the crystalline structures of the SLG were partly damaged after cationization. In addition, two new types of CSLG-based nanocomplex were developed by interacting with LMP and HMP through electrostatic interactions to obtain the LMP-CSLG and HMP-CSLG nanocomplex particles, respectively. The morphology of the nanocomplexes all showed a regular spherical structure and monodispersity with a particle size <700 nm. The increased CSLG concentration enhanced the interactions between the two biopolymers of the nanocomplex. The nanocomplex particles were stable in the simulated GI fluid, and the stability was further improved with the increase in the CSLG concentration. In addition, the HMP-CSLG have a higher encapsulation efficiency for the TAN compared with the LMP-CSLG nanocomplex. Furthermore, the TAN-loaded nanocomplexes provided control and sustained release of TAN in the simulated GI fluid. These findings suggested that the CSLG-based nanocomplexes hold promise as oral delivery for some active ingredients.

## Figures and Tables

**Figure 1 foods-10-02509-f001:**
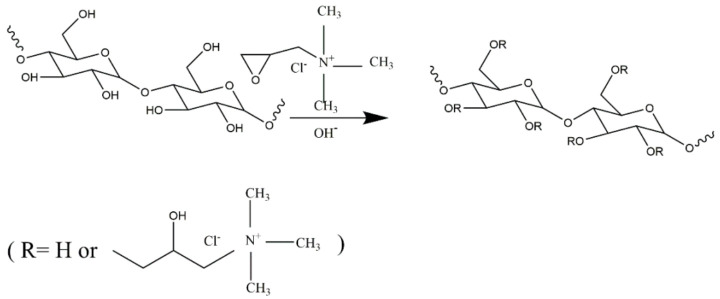
Schematic synthesis of cationic short linear glucan.

**Figure 2 foods-10-02509-f002:**
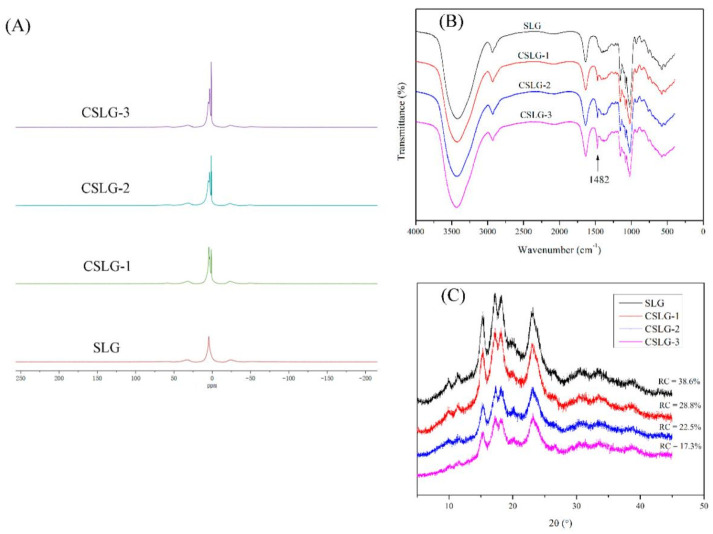
(**A**) the ^1^H NMR, (**B**) FTIR, and (**C**) X-ray spectra of SLG and CSLG; Proton nuclear magnetic resonance (^1^H NMR); cationic short linear glucan (CSLG) 1 mL; 2 mL; 3 mL; short linear glucan (SLG).

**Figure 3 foods-10-02509-f003:**
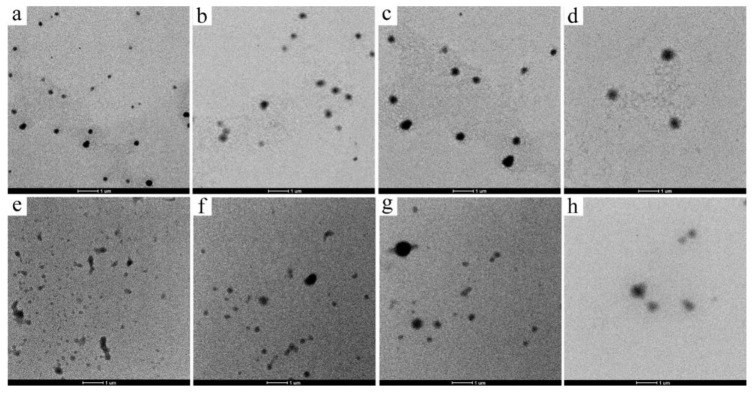
TEM images for morphological observation. (**a**–**d**) is the TEM images of LMP-CSLG nanocomplex particle with different CSLG concentration (0.5, 1.0, 2.5, and 5.0 mg/mL); (**e**–**h**) is the TEM images of HMP-CSLG nanocomplex particle with different CSLG concentration. Transmission Electron Microscopy (TEM); low methoxyl pectin (LMP); cationic short linear glucan (CSLG); high methoxyl pectin (HMP).

**Figure 4 foods-10-02509-f004:**
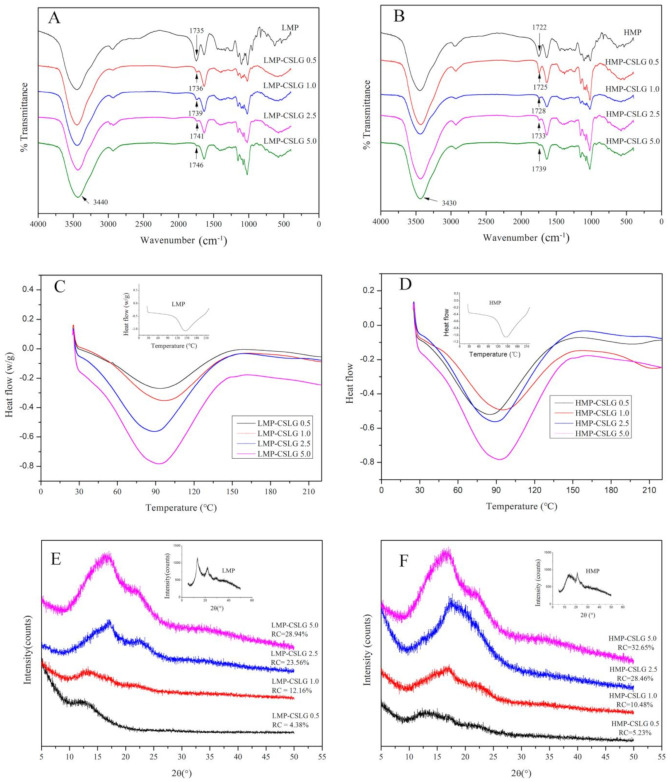
(**A**) FTIR spectra of LMP and LMP-CSLG nanocomplex with different CSLG concentration (0.5, 1.0, 2.5, and 5 mg/mL) (**B**) FTIR spectra of HMP and HMP-CSLG nanocomplex with different CSLG concentration; (**C**) The DSC curves of LMP and LMP-CSLG nanocomplex with different CSLG concentration; (**D**) HMP and HMP-CSLG nanocomplex with different CSLG concentration; (**E**) X-ray spectra of LMP and LMP-CSLG nanocomplex with different CSLG concentration; (**F**) X-ray spectra of HMP and HMP-CSLG nanocomplex with different CSLG concentration. Fourier transform infrared spectroscopy (FTIR); Differential Scanning Calorimetry (DSC); low methoxyl pectin (LMP); Cationic short linear glucan (CSLG); high methoxyl pectin (HMP).

**Figure 5 foods-10-02509-f005:**
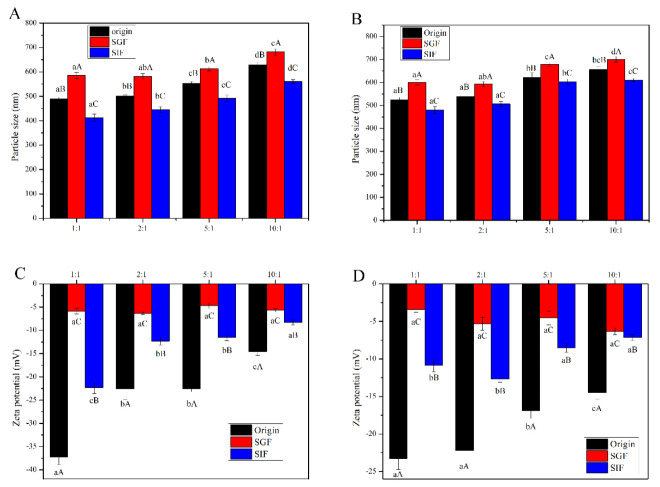
Particle size of (**A**) LMP-CSLG and (**B**) HMP-CSLG nanocomplexes with different CSLG concentration under simulated gastric and intestinal fluids. Zeta potential of (**C**) LMP-CSLG and (**D**) HMP-CSLG nanocomplexes with different CSLG concentration under simulated gastric and intestinal fluids. ^a–d^ Values followed by different letters in the same column are significantly different (*p* < 0.05), where a is the lowest value. ^A–C^ Values followed by different letters in the Scheme 0. where A is the highest value. Low methoxyl pectin (LMP); Cationic short linear glucan (CSLG); high methoxyl pectin (HMP).

**Figure 6 foods-10-02509-f006:**
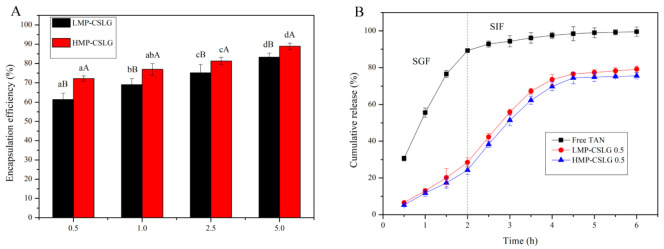
(**A**) Encapsulation efficiency of TAN in CSLG-based nanocomplex particles with different CSLG concentration (0.5, 1.0, 2.5, and 5 mg/mL), and (**B**) TAN release from the LMP-CSLG and HMP-CSLG nanocomplex particles with a CSLG concentration of 0.5 mg/mL under simulated gastrointestinal fluids. Low methoxyl pectin (LMP); Cationic short linear glucan (CSLG); high methoxyl pectin (HMP); Tangeretin (TAN). ^a–d^ Values followed by different letters in the same column are significantly different (*p* < 0.05), where a is the lowest value. ^A,B^ Values followed by different letters in the Scheme 0. where A is the highest value.

**Table 1 foods-10-02509-t001:** Degree of substitution and zeta potential of SLG and CSLG.

Sample	DS	Zeta Potential
SLG	0	−12.13 ± 0.16
CSLG-1	0.36 ± 0.03	22.18 ± 1.48
CSLG-2	0.58 ± 0.01	30.54 ± 1.08
CSLG-3	0.94 ± 0.04	34.52 ± 1.38

Cationic short linear glucan (CSLG) 1 mL; 2 mL; 3 mL; short linear glucan (SLG); degree of substitution (DS).

**Table 2 foods-10-02509-t002:** Particle size, PDI, and zeta potential of LMP-CSLG and HMP-CSLG complex nanoparticles.

Sample	Particle Size	PDI	Zeta Potential
LMP-CSLG			
LMP-CSLG 0.5	488.8 ± 6.61 ^a^	0.203 ± 0.01 ^a^	−37.28 ± 1.64 ^a^
LMP-CSLG 1.0	500.9 ± 5.16 ^a^	0.253 ± 0.01 ^b^	−22.58 ± 2.36 ^b^
LMP-CSLG 2.5	553.5 ± 6.55 ^b^	0.286 ± 0.02 ^b^	−22.52 ± 0.68 ^b^
LMP-CSLG 5.0	629.2 ± 11.55 ^c^	0.300 ± 0.01 ^c^	−14.53 ± 0.85 ^c^
HMP-CSLG			
HMP-CSLG 0.5	524.0 ± 10.85 ^a^	0.253 ± 0.02 ^a^	−23.28 ± 1.46 ^a^
HMP-CSLG 1.0	539.2 ± 14.29 ^a^	0.276 ± 0.01 ^a^	−22.21 ± 1.03 ^a^
HMP-CSLG 2.5	621.3 ± 20.85 ^b^	0.326 ± 0.01 ^b^	−16.91 ± 1.03 ^b^
HMP-CSLG 5.0	656.2 ± 13.81 ^c^	0.386 ± 0.02 ^c^	−14.5 ± 0.84 ^c^

^a–c^ Values followed by different letters in the same column are significantly different (*p* < 0.05), where a is the lowest value; low methoxyl pectin (LMP); Cationic short linear glucan (CSLG); high methoxyl pectin (HMP); Polymer Sispersity Index (PDI).

## Data Availability

The data used to support the findings of this study are available from the corresponding author upon request.

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
