# Peer review of "Characterization of Cationic Modified Short Linear Glucan and Fabrication of Complex Nanoparticles with Low and High Methoxy Pectin"

_foods, 2021, doi:10.3390/foods10102509_

Round 1

Reviewer 1 Report

The authors reported on the chemical modification of short linear glucan aiming at introducing cationic charges and its further use for the preparation complex nanoparticles through electrostatic iteractions and H bonding with methoxyl pectin. The so-obtained nanoparticles were used for the encapsulation of tangeretin. The potential as oral delivery system was evaluated through investigating encapsulation efficiency, digestive stability and release profiles. The MS lacks of originality and comparison with similar published studies should have been provided. The MS needs to be carefully proofread by a native English speaker as several grammatical mistakes remain.  

More troubling, somme of the presented results are difficult to be understood in the present form. 

The chemical structure of the SLG after the chemical modification should be included if not the chemical reaction path. To what refers CSGL-1,2,3? Is the unit in Figure 3A nm? To what corresponds the x-axis in Figure 5A? Where are Figures 5C and D?  

Size values such as 585.26 nm should not be given as it does not sound scientifically relaible. Can you measure 0.01 nm? Table 2 does  seem not to have caption. 

The particle size given in the table does not seem to agree well with TEM images. 

Author Response

Review

  1. The authors reported on the chemical modification of short linear glucan aiming at introducing cationic charges and its further use for the preparation complex nanoparticles through electrostatic iteractions and H bonding with methoxyl pectin. The so-obtained nanoparticles were used for the encapsulation of tangeretin. The potential as oral delivery system was evaluated through investigating encapsulation efficiency, digestive stability and release profiles. The MS lacks of originality and comparison with similar published studies should have been provided. The MS needs to be carefully proofread by a native English speaker as several grammatical mistakes remain.

Response:

Thank you for your suggestions. We have added some comparison with the similar published studies in the MS.

Line 209-211: “Xu et al [25]. found that the zeta potentials of quaternized β-chitin derivatives in-creased from +25.4 to +44.1 mV with the degree of quaternization increasing from 0.18 to 0.43.”

Line 363-364: “Chang et al [40]. has demonstrated that pectin coating provided excellent stability to NaCas/zein nanoparticles under simulated gastrointestinal conditions.”

Line 414-416: “Liu et al [44] has been reported that for both cationic starch/κ-carrageenan and cati-onic starch/Carboxymethyl chitosan nanoparticles, the release rate of EGCG in SIF was slightly higher than that in SGF.”

We have carefully checked revised the syntax and grammar throughout the MS, and the English language of the MS has been professionally edited by a native English speaker.

  1. More troubling, some of the presented results are difficult to be understood in the present form.

The chemical structure of the SLG after the chemical modification should be included if not the chemical reaction path. To what refers CSLG-1,2,3? Is the unit in Figure 3A nm? To what corresponds the x-axis in Figure 5A? Where are Figures 5C and D? 

Response:

Thank you for your question.

Line 111: The chemical structure of the CSLG and the chemical reaction path of cationic modification was showed in Fig. 1.

Fig. 1 Schematic synthesis of Cationic short linear glucan

Line 110-111: “The SLG modified by CHPTAC with different content (1mL, 2mL and 3mL) was named as CSLG-1, CSLG-2 and CSLG-3 respectively.”

I am very sorry for my carelessness in Fig. 4A and B, and the unit of the wavenumber has been revised.

Figure 6: (A) Encapsulation efficiency of TAN in CSLG-based nanocomplex particles with different CSLG concentration (0.5, 1.0, 2.5 and 5 mg/mL).

I am sorry for my carelessness about the order of the figure in the MS, and we have carefully checked and revised throughout the MS.

  1. Size values such as 585.26 nm should not be given as it does not sound scientifically reliable. Can you measure 0.01 nm? Table 2 does seem not to have caption.

Response:

Thank you for your comments. The value of the particle size in table 2 was revised, and only one digit after the decimal point is retained. 

The caption of the Table 2 has been added in the MS.

  1. The particle size given in the table does not seem to agree well with TEM images.

Response:

Line 276-278: “This is because the particle size from TEM is obtained from dehydrated particles, that have shrinkage in the dimension at their dry status under TEM observation.”

Reviewer 2 Report

1. Use of abbreviation should be revised. For example, the authors named CDBS but CSLG was used. Abbreviation should be shown at the first place in the manuscript. What does PDI stand for? 2. Table 1 has unnecessary numbers. The legend of Table 2 locates the inappropriate place. 3. Figures should be revised. For example, in Figure 3, some are in the text next to lines, some are in the legend form. Use same color and text for same samples. 4. Reference format should be identical. For example, reference format in line 310 should be changed.

Author Response

Review

Comments and Suggestions for Authors

  1. Use of abbreviation should be revised. For example, the authors named CDBS but CSLG was used. Abbreviation should be shown at the first place in the manuscript. What does PDI stand for? 2. Table 1 has unnecessary numbers. The legend of Table 2 locates the inappropriate place. 3. Figures should be revised. For example, in Figure 3, some are in the text next to lines, some are in the legend form. Use same color and text for same samples. 4. Reference format should be identical. For example, reference format in line 310 should be changed.

Response:

Thank you for your comments. I am very sorry for my carelessness and we have revised it.

Line 110: “……named as CSLG-1, CSLG-2 and CSLG-3 respectively.”

Line 147: “Particle size, Polymer dispersity index (PDI) and zeta potential”

The unnecessary numbers in Table 1 has been deleted.

The legend of Table 2 has been revised.

The Figure 3 has been carefully checked and revised. The same samples were marked with the same color.

The reference format has been carefully checked and revised the reference format to make all reference format identical.

Line 320: “The smooth thermograms indicate the inherent amorphous structures of native bi-opolymer and the binary complexes [33]”

Round 2

Reviewer 1 Report

The authors have improved raisonnably the overall quality of their MS.